# Care Bundles to Improve Hemoperfusion Performance in Patients with Severe COVID-19: A Retrospective Study

**DOI:** 10.3390/jcm13123360

**Published:** 2024-06-07

**Authors:** Sirirat Mueankwan, Konlawij Trongtrakul, Pattraporn Tajarernmuang, Nutchanok Niyatiwatchanchai, Prit Kusirisin, Phoom Narongkiatikhun

**Affiliations:** 1Surgical Intensive Care Unit, Division of Nursing Services, Faculty of Medicine, Chiang Mai University, Chiang Mai 50200, Thailand; sirirat.m@cmu.ac.th; 2Division of Pulmonary, Critical Care, and Allergy, Department of Internal Medicine, Faculty of Medicine, Chiang Mai University, Chiang Mai 50200, Thailand; pattraporn.t@cmu.ac.th (P.T.); nutchanok.n@cmu.ac.th (N.N.); 3Division of Nephrology, Department of Internal Medicine, Faculty of Medicine, Chiang Mai University, Chiang Mai 50200, Thailand; jingprit@hotmail.com (P.K.); phoom.n@cmu.ac.th (P.N.)

**Keywords:** care bundles, critical illness, COVID-19, cytokine storms, HA330, hemoperfusion

## Abstract

**Background/Objectives:** Hemoperfusion (HP) is employed to modulate cytokine storms in severe coronavirus disease 2019 (COVID-19) patients, requiring careful attention for success and safety. Therefore, we investigated whether our care bundles could enhance HP performance. **Methods:** We conducted a retrospective cohort study on adult patients (≥20 years old) with severe COVID-19 pneumonia. In the first wave (Phase I), we identified HP-related issues and addressed them with care bundles in the second wave (Phase II). The care bundles included early temperature control, precise hemodynamic monitoring, and clot prevention measures for the HP membrane. The HP success rate and associated adverse events (AEs) were assessed between the two phases. **Results:** The study included 60 HP (HA330) sessions from 27 cases (Phase I: 21 sessions from 9 cases; Phase II: 39 sessions from 18 cases). Patient characteristics and treatments for COVID-19 were similar, except for baseline body temperature (BT) and heart rate (HR). Phase II showed a higher success rate (67% vs. 89%, *p* = 0.19), although it did not reach statistical significance. Phase I recorded a significantly higher frequency of AEs (3 [IQR 1, 4] events/case vs. 1 [IQR 0, 2] events/case, *p* = 0.014). After implementing the care bundles, hypothermia significantly decreased (78% vs. 33%, *p* = 0.037), with an adjusted odds ratio of 0.15; 95% CI 0.02–0.95, *p* = 0.044 for baseline BT. **Conclusions:** Further exploration with a larger sample size is required to establish the advantages of care bundles. However, the bundles’ implementation has significantly improved hypothermia prevention.

## 1. Introduction

The coronavirus disease 2019 (COVID-19) pandemic ushered the world into a chaotic situation [1,2]. The emergence of this new contagious disease intensified COVID-19 into becoming a burdensome disease. Numerous countries experienced a surge in the number of patients, resulting in a high rate of morbidity and mortality [3,4,5,6].

Excessive production of inflammatory cytokines, such as interleukin-1 (IL-1), interleukin-6 (IL-6), and tumor necrotic factor alpha (TNF-alpha), significantly contributes to multi-organ dysfunction in COVID-19 patients [7,8,9,10]. Modulating the inflammatory response with treatments such as corticosteroid [11,12], IL-6 receptor antagonist [13,14], and hemoperfusion (HP) [15,16] is crucial for promoting survival. However, all of these treatments have distinct indications and levels of effectiveness [7,8,9].

HP serves as a rescue therapy for severe COVID-19, countering cytokine storms caused by the virus [17,18,19]. However, it demands an expert team and carries the risk of peri-procedural adverse events (AEs). During the first pandemic (Phase I), problems related to HP care were addressed in subsequent phases (Phase II) by implementing a care bundle, which aided nurses in conducting HP effectively. We investigated HP success rates and associated AEs before and after implementing care bundles in severe COVID-19 patients.

## 2. Materials and Methods

### 2.1. Study Design

We conducted a retrospective observational study using the data from COVID-19 patients treated during the period from 12 April 2021 to 27 January 2022 (Phase I: 12 April 2021 to 19 May 2021, and Phase II: 14 July 2021 to 27 January 2022, with 63 and 180 patients, respectively).

The study received approval from the Research Ethics Committee of the Faculty of Medicine, Chiang Mai University, Thailand (study code: NUR-2566-0444). The study was performed following the Declaration of Helsinki, which outlines ethical principles for medical research involving human subjects. An informed consent waiver was approved by the Research Ethics Committee of the Faculty of Medicine, Chiang Mai University, due to the minimal risk and anonymous data analysis intrinsic to the retrospective manner of this study.

We retrieved the information from those patients with severe COVID-19 who were admitted to the ten-bed intensive care unit (ICU) of the Chiang Mai Neurological Hospital, which was under a joint memorandum operation by the staff from the Faculty of Medicine of Chiang Mai University.

### 2.2. Inclusion and Exclusion

The participants were considered eligible if they were adults aged 20 or older; were hospitalized with severe COVID-19 pneumonia, as defined by a score from six to nine on the World Health Organization (WHO)’s ordinal scale [20]; and received treatment with high-flow nasal cannula (HFNC), non-invasive ventilation (NIV), or invasive mechanical ventilation (IMV). They also needed to be confirmed positive for severe acute respiratory syndrome-coronavirus-2 (SARS-CoV-2) through a reverse transcription polymerase chain reaction (RT-PCR) test on respiratory tract specimens, present evidence of pulmonary infiltration on chest X-ray images, and undergo HP treatment during their ICU stay.

Exclusion criteria included patients who consented to limited treatment and advanced directives for medical therapy.

### 2.3. Definitions of Severe COVID-19 Pneumonia

Patients diagnosed with severe COVID-19 pneumonia were defined according to a score from six to nine on the ordinal scale created by the WHO Working Group on the Clinical Characterization and Management of COVID-19 Infection [17]. This included patients who were hospitalized and utilized (1) high-flow nasal cannula (HFNC) or non-invasive ventilation (NIV); (2) intubation and mechanical ventilation (IMV) with PaO_2_/FiO_2_ ratio (PF ratio) ≥ 150 or SpO_2_/FiO_2_ ≥ 200; (3) IMV with PF ratio < 150 (SpO_2_/FiO_2_ < 200) or vasopressors therapy; and (4) IMV with PF ratio < 150 and vasopressor, dialysis, or extracorporeal membrane oxygenation (ECMO) [17].

### 2.4. Data Collection

The data were retrieved from the Electronic Medical Records by the first author, SM. The information included the patients’ demographics, pre-existing comorbidities, vital signs, severity of illness, and laboratory test results at the initial phase of HP therapy. Throughout each patient’s ICU stay, we documented treatments and types of respiratory support.

The ICU mortality rate, hospital mortality rate, ICU length of stay, and hospital length of stay were examined. Furthermore, we reviewed HP-related information, including the initiation date of HP relative to the admission date, the total number of HP sessions, the total duration of HP operation, the success rate of HP, and HP-related AEs.

### 2.5. Study Outcomes

The primary outcome was to determine the success rate of HP before and after the implementation of care bundles, defined as the achievement of a completed four-hour HP session without experiencing interruptions that significantly affected the HP procedure.

Other outcomes related to HP-associated AEs, including shivering, cardiac arrhythmia, hypotension, hypertension, hypothermia, and cartridge clotting, were compared between the phases.

### 2.6. Standard of Care for Severe COVID-19 Patients

All patients received oral favipiravir or intravenous remdesivir, depending on the severity of hospitalization. All patients were switched to remdesivir if the disease was determined to be progressing. Concomitant intravenous systemic corticosteroid, primarily dexamethasone, or an equivalent dose of hydrocortisone, methylprednisolone, or prednisolone was given as determined by the attending physician. Tocilizumab was optionally provided in severe cases. Empirical antibiotics were administered if indicated. Respiratory support was offered using HFNC, NIV, or IMV, as appropriate based on the patient’s respiratory status.

### 2.7. Hemoperfusion Setting and Prescription

HP was conducted during the early stages of hospitalization in patients with progressive acute hypoxemic respiratory failure (AHRF) with a PiO_2_/FiO_2_ < 200, positive evidence of systemic hyperinflammation (lymphopenia < 1000 cells/µL or a high level of CRP > 30 mg/L), despite receiving standard therapy for COVID-19. The utilization of HP was determined based on the attending intensivist and nephrologist and was performed by the ICU nurses after proper establishment of vascular access using an 11.5 Fr double-lumen catheter.

The hemoadsorption HA330 cartridge (Jafron Biomedical, Zhuhai, China), integrated with the HP machine, was utilized in our center. We primed the cartridge with 5000 IU unfractionated heparin (UFH) for 30 min. Then, we rinsed the cartridge with 0.9% normal saline for 4 L at a flow rate of 100 mL/min. Since most patients had already received low-molecular weight heparin, a standard prophylactic therapy for hospitalized COVID-19 patients, no additional UFH was administered during the HP session.

We set the HP temperature at 37.0 °C and initiated the blood flow rate (Qb) at 80 mL/min. We gradually increased the Qb to achieve 150 to 200 mL/min within ten minutes. We recommended a four-hour period of HP for at least two sessions, 24 h apart. However, one to four sessions might have been merited, depending on the patient’s severity.

### 2.8. Care Bundles for Hemoperfusion

HP care-related issues discovered in the first wave of the pandemic (Phase I) were addressed during the second wave (Phase II) by specialized ICU nurses trained in renal replacement therapy. The AEs in Phase I included shivering, cardiac arrhythmia, hypotension, hypertension, hypothermia, cartridge clotting, and circuit shattering. These AEs were subsequently addressed, and care bundles were implemented to assist nurses in performing HP. The strategies to promote a four-hour HP session included early temperature control, precise hemodynamic monitoring with early management, and measures for clot prevention in the HP membrane. We employed regular monitoring of each patient’s vital signs before the HP procedure and at 5, 15, 30, 45, 60, 120, 180, and 240 min thereafter. Additionally, nurses actively monitored for AEs and promptly engaged physicians for necessary management, as shown in Table 1.

### 2.9. Statistical Analysis

Continuous data were summarized as median and interquartile ranges (IQR). Categorical variables were summarized as numbers and percentages. For the comparison of continuous variables, the Wilcoxon rank-sum test was employed, while for categorical variables, Fisher’s exact test was utilized.

Univariable logistic regression analysis was utilized to illustrate the association of the care bundles, as the independent variable, with the dependent outcome. The dependent variables were put into the model one by one in a binary form (yes/no), which included the success rate and each AE. The results were reported as the odds ratio (OR) and 95% confidence interval (95%CI).

Given the constraints of the limited sample size, we focused our selection on pertinent variables that exhibited differences at the baseline and were related to the outcome of interest for refining the regression model, such as adjusting body temperature in the model where the outcome was hypothermia.

A significant *p*-value of less than 0.05 was considered statistically significant. For analyzing data, we utilized STATA version 16.0 (Stata Corp LP, College Station, TX, USA).

## 3. Results

### 3.1. Demographics, Clinical Features, Treatment, and Outcomes

We included all patients with severe COVID-19 who underwent HP at our center. Twenty-seven cases were involved in the study (Phase I: n = 9 and Phase II: n = 18). Table 2 summarizes the baseline characteristics of the patients. When comparing Phase I and Phase II, there were no significant differences in patients’ demographics, including age and gender, with median values of 63 [IQR 53, 67] years vs. 58 [IQR 52, 67] years, *p* = 0.81, and females comprising 56% vs. 28%, *p* = 0.22, respectively.

Moreover, pre-existing comorbidities and vital signs were also comparable between phases, except for body temperature (BT) and heart rate (HR). Both BT and HR were significantly lower in Phase I than in Phase II, with the values of 36.0 [IQR 35.6, 36.4] °C vs. 36.7 [IQR 36.4, 37.2] °C, *p* < 0.001, and 62 [IQR 52, 76] bpm vs. 80 [IQR 72, 88] bpm, *p* = 0.044, respectively.

Additionally, there were no discernible variations observed in the results of laboratory investigations, including absolute lymphocyte count, D-dimer, C-reactive protein, and interleukine-6, which were not different between phases (all *p* > 0.005). Table 2 provides further information about the treatments administered during ICU admission. Of note, there were no significant differences in the treatments, types of respiratory support, and patient outcomes between the two phases (all *p* > 0.05).

### 3.2. HP-Related Information and Outcomes

All HP-related information is summarized in Table 3. The time to first HP initiation from admission was not different between phases (1 [IQR 1, 2] day vs. 3 [IQR 2, 6] days, *p* = 0.08). In total, there were 60 HP sessions/27 cases (Phase I: 21 sessions/9 cases and Phase II: 39 sessions/18 cases), with no significant differences in the median HP sessions per case or the total duration of HP operation between phases (2 [IQR 2, 3] sessions vs. 2 [IQR 2, 3] sessions, *p* = 0.78, and 480 [IQR 420, 720] minutes vs. 480 [IQR 480, 720] minutes, *p* = 0.91).

The success rate of HP per case, calculated from the number of cases that totally completed a 4 h HP session divided by the number of cases from each phase, was slightly higher in Phase II (Table 3). However, it did not reach statistical significance (67% vs. 89%, *p* = 0.19). The success rate remained the same when considered per session, calculated from the number of completed 4 h sessions divided by the number of sessions from each phase (81% vs. 95%, *p* = 0.11).

The total number of AEs was 49 events (Phase I: 26 events and Phase II: 23 events). The median number of AEs per case was significantly higher in Phase I than in Phase II (3 [IQR 1, 4] events/case vs. 1 [IQR 0, 2] events/case, *p* = 0.014). The details regarding the number of AEs per case differed between phases (*p* = 0.039), as summarized in Table 3.

While the incidences of shivering, cardiac arrhythmia, hypotension, and hypertension were not significantly different between phases (all *p* > 0.05), hypothermia exhibited a statistically significant reduction (78% vs. 33%, *p* = 0.037), with an OR of 0.14; 95% CI 0.02–0.91, *p* = 0.039 (Figure 1). Adjusting for different baseline BT values, the risk of hypothermia remained significantly reduced in Phase II, with an OR of 0.15; 95% CI 0.02–0.95, *p* = 0.044 (Figure 1).

## 4. Discussion

Our study offers insights drawn from the first wave of the pandemic that were then subsequently applied to the second wave of the pandemic, involving cytokine reduction using HP with HA330 for severe COVID-19 patients. We incorporated several technical aspects to improve HP performance, including early temperature control, regular hemodynamic monitoring, ongoing surveillance for AEs, and timely contact with physicians to provide essential interventions when necessary. Although our approach did not lead to a significantly greater success rate of HP, it significantly reduced the number of AEs, particularly the incidence of hypothermia.

Interestingly, there was a slightly higher number of patients undergoing HP therapy in Phase I compared to Phase II, though the amount lacks statistical significance. The proportion of patients receiving HP during these two phases was 14.3% (9/63 cases) vs. 10.0% (18/180 cases), respectively, with a *p*-value of 0.36. One contributing factor to the reduction in the utilization of HP treatment in Phase II pertained to the necessity of vascular access and specialized nurse support for continuous bedside HP operations lasting at least four hours. This additional complexity rendered HP more intricate than mere medication administration.

Moreover, as time progressed, more knowledge supporting best practices for treating COVID-19 patients continued to evolve. We noticed some disparities in patient treatment between the two phases. The administration of systemic corticosteroids showed a tendency towards a longer duration in Phase II (6 [IQR 5, 8] days vs. 12 [IQR 9, 16] days, *p* = 0.06). Additionally, the usage of Tocilizumab, an IL-6 receptor inhibitor, was more prevalent in Phase II (11% vs. 44%, *p* = 0.09).

Nonetheless, there were a total of 60 HP sessions in our study (Phase I: 21 sessions and Phase II: 39 sessions). The comparative success rate of HP between Phase I and Phase II showed no statistical difference when adjusted by case or by sessions (67% vs. 89%, *p* = 0.19 and 81% vs. 95%, *p* = 0.11, respectively). Remarkably, the introduction of the HP care bundles led to a substantial decrease in the median number of AEs (3 [IQR 1, 4] events/case vs. 1 [IQR 0, 2] event/case, *p* = 0.014). While shivering, cardiac arrhythmia, hypotension, and hypertension did not exhibit significant differences, hypothermia demonstrated a statistically significant reduction (78% vs. 33%, *p* = 0.037), with an OR of 0.15; 95% CI 0.02–0.95, *p* = 0.044 when adjusted for baseline BT. This result implies that the implementation of the care bundle during HP effectively mitigated the occurrence of hypothermia.

Although the ICU mortality rate was marginally lower in Phase II, there was no statistically significant difference between the two phases (33% vs. 28%, *p* = 0.55). One plausible explanation could be the slightly delayed initiation of HP compared to Phase I, with the median time to first HP initiation from ICU admission being 1 (IQR 1, 2) day vs. 3 (IQR 2, 6) days, *p* = 0.008. A delay in HP initiation could potentially impact the efficacy of HP. Exploring the effectiveness of cytokine modulation through a combination of early HP and cytokine reduction medication could offer valuable insights for future research. On the other hand, investigating the efficacy by comparing medication alone to early HP could also present another area of interest.

It is well-established that cytokine storms contribute to endothelial dysfunction, trigger microvascular thrombosis, and lead to organ dysfunction such as acute respiratory distress and acute kidney injury [21]. Consequently, the cytokine storm is a major contributor to an increased mortality rate among critically ill patients with COVID-19 [22]. Therefore, a strategic intervention aimed at the timely and effective clearance of cytokines through HP may lead to improved outcomes for severe COVID-19 patients [10,17,18,19].

Several studies have explored the application of HP in severe COVID-19 patients, utilizing various cartridge types, including HA330 [17,19], HA380 [23], HA230 in combination with HA280 [24], CytoSorb [25,26,27], and oXiris [28,29]. While HP has shown promise in improving SpO_2_ levels and reducing inflammatory cytokines, it is important to note that most of these studies have been constrained by their small sample sizes and case series designs [23,24,25,26,28,29].

One of the largest investigations of CytoSorb involved the CYCOV trial [27], which revealed that the level of IL-6 at 72 h after ECMO initiation did not significantly differ between the HP + ECMO group (n = 17) vs. the ECMO alone group (n = 17). However, the 30-day survival rate was significantly lower in the HP + ECMO group (18% vs. 76%, *p* = 0.002). Caution is required when considering HP during the early phase of ECMO therapy, as HP can also remove anti-inflammatory cytokines, which might potentially contribute to this adverse outcome.

To our knowledge, one large retrospective study (n = 128) compared the use of HP with HA330 (n = 46) and CytoSorb (n = 9) to matched controlled patients (n = 73) [19]. This study showed a lower ICU mortality in the HP group than in the control group (67% vs. 89%, *p* = 0.002) [19]. The HP group also exhibited more favorable outcomes in terms of a shorter ICU length of stay, greater improvements in SpO_2_, and greater reductions in PaCO_2_ when compared to the control group [19]. However, it is worth noting that the mortality rate in this study was somewhat higher than in our study (30%). Nonetheless, intubation rates were similar, with approximately three quarters of patients being intubated.

Another study employing HA330 was conducted by Surasit K et al., where HP was performed for three or more sessions (n = 15) compared to no HP or less than three sessions of HP (n = 14) [17]. The results indicated that patients who received three or more HP sessions achieved a reduction in organ dysfunction (as measured by the SOFA score), a decrease in pulmonary infiltration appearing on chest X-ray images, and lower CRP levels. Furthermore, the HP group outperformed the control group in terms of ICU mortality and 28-day mortality (13.3% vs. 92.9%, *p* < 0.001 and 13% vs. 86%, *p* < 0.001, respectively).

Although several studies have demonstrated the advantages of HP for severe COVID-19 patients [17,19,23,24,25,26,27,28,29], a notable deficiency exists in the available evidence to support best practices for enhancing HP performance. In addition, the overwhelming number of patients with severe COVID-19 led to a widespread shortage of healthcare professionals. In response, some centers eventually enlisted multidisciplinary healthcare workers who may not be specialized in critical care medicine. Our center encountered this challenge as well. Therefore, the execution of HP operations could occasionally present challenges. The guidance from experienced nurses, coupled with comprehensive care bundles encompassing all essential elements, became indispensable. It is worth noting that this study originates from a resource-limited setting where the implementation of HP can be somewhat financially burdensome. Given that the success rate of HP operations could potentially influence the survival of severe COVID-19 patients, it becomes imperative to develop a strategy for enhancing HP performance.

Hypothermia, or suboptimal thermal regulation, represented a noteworthy challenge in the context of extracorporeal organ support. Studies have indicated that hypothermia affects nearly half of all patients undergoing continuous renal replacement therapy (CRRT) [30,31]. This condition demands attention due to its potential to exacerbate patients’ thermal instability, thereby heightening susceptibility to sepsis; precipitate chills; and induce arrhythmias and hemodynamic instability. Furthermore, individuals encountering hypothermia during CRRT were confronted with a notable increase in mortality risk, with rates reaching up to 60% [31]. It is anticipated that the incidence of hypothermia will remain consistent for HP. Although certain centers may exhibit a heightened susceptibility to hypothermia, it is important to note that some HP machines, including those in our center, lack integrated warming capabilities. Therefore, a protocol to monitor patients’ body temperatures is necessitated to prevent this complication.

We conducted frequent monitoring of each patient’s vital signs before the HP procedure and at intervals of 5, 15, 30, 45, 60, 120, 180, and 240 min thereafter. This standardized monitoring of vital signs before and after the initiation of HP was not fully established in Phase I. There was an understanding that vital signs were observed every 15 min during the initial hour of HP, and subsequently every 60 min, in accordance with the tailored nursing approaches of each unit. Nevertheless, to foster uniformity in nursing practices, efforts were made to integrate these practices into a formalized protocol or agreement, particularly in Phase II. Regular monitoring of vital signs prompted nursing action during HP. Furthermore, the comprehensive bundles facilitated timely physician consultation in cases where initial management falls short.

An additional issue that impacted the effectiveness of HP was the formation of blood clots within the membrane. Furthermore, blood loss of approximately 285 mL (HA330 = 185 mL and circuit = 100 mL) may occur during circuit disposal. Thus, it is advisable to promptly address any early indications of membrane clotting. This can be achieved by regularly checking vascular access, adjusting patient positioning, closely monitoring for any increase in transmembrane pressure (TMP), and further examining clot size when TMP rises. In advanced stages where machine operation is compromised due to clotting, a more vigorous blood return is recommended in order to minimize blood loss whenever feasible.

Our study had some limitations. Firstly, the small number of patients suitable for the investigation into the study hypothesis—centered on the care bundles to enhance HP performance—impeded a comprehensive assessment of HP benefits, including the success rates and other efficacies such as disease progression, intubation, and mortality rates. Further investigations with larger sample sizes could provide more insights to assess the impact of the care bundles, in cooperation with standardized treatment protocol, including the type and duration of corticosteroid treatment or the use of IL-6 receptor inhibitor, on patients’ survival outcomes. However, in terms of efficacy we observed a significant reduction in IL-6, an important surrogate marker for disease severity, following HP treatment. In all phases, the IL-6 levels decreased from 75 (IQR 29, 109) pg/mL to 25 (IQR 9, 60) pg/mL, *p* < 0.001. This reduction was consistent across both phases. In Phase I, IL-6 levels dropped from 79 (IQR 39, 86) pg/mL to 35 (IQR 9, 60) pg/mL, *p* < 0.005. In Phase II, IL-6 levels dropped from 68 (IQR 29, 109) pg/mL to 25 (IQR 13, 27) pg/mL, *p* = 0.03. Secondly, the improved HP performance in Phase II might be attributed to the experience gained during Phase I. Although experienced nurses managed the care bundles, it was not always feasible to constantly have specialized nurses available. Therefore, the designated nurses were justified in following these care bundles. Thirdly, the retrospective nature of our study may introduce biases due to missing data. Regrettably, during the study period, our facility encountered constraints in conducting cytokine analyses, particularly for IL-6, during Phase II. Hence, we offer the available data, recognizing certain limitations. Nevertheless, we observe that the initial IL-6 levels in both phases exhibit comparability. Nevertheless, we posit that certain surrogates could reasonably approximate cytokine storms or disease severity, such as absolute lymphocyte count, D-dimer, and C-reactive protein levels. Moreover, it appears that IL-6 levels themselves may not have a direct impact on the success or failure of HP operations. Lastly, we could not determine how the care bundles affected COVID-19 patients versus those with septic shock. We occasionally perform HP on septic shock patients, but we do not have enough data for the comparison.

## 5. Conclusions

Further investigation with a larger sample size is necessary to confirm the benefits of the care bundles. Nonetheless, the utilization of these care bundles has been shown to notably improve the safety of HP, with a specific focus on the successful prevention of hypothermia.

## Figures and Tables

**Figure 1 jcm-13-03360-f001:**
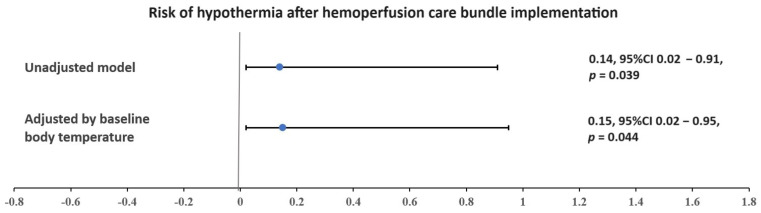
Risk of hypothermia after hemoperfusion care bundle implementation. Blue dots and horizontal lines with bars indicate odds ratios and 95% confidence intervals, respectively.

**Table 1 jcm-13-03360-t001:** Adverse events and care bundles for hemoperfusion.

Adverse Events	Definition	Care Bundles
Shivering	The presence of body shaking or teeth chattering	Give paracetamol, opioids, or sedative Check for membrane allergy Provide warm blanket if hypothermia
Cardiac arrhythmia	New onset of atrial fibrillation/flutter, premature ventilation, among others	Ask about chest pain Check volume status Check electrolytes Reduce blood flow rate Give anti-arrhythmic drugs
Hypotension	Mean arterial pressure ≤ 65 mmHg	Decrease preset temp to 36.5 °C Check patient volume status Fluid challenge, if indicated Reduce blood flow rate
Hypertension	Blood pressure > 140/90 mmHg	Manage pain/agitation Give anti-hypertensive drugs
Hypothermia	Body temperature < 36.0 °C	Check patient temp before HP Increase HP preset temp Provide warm blanket
Cartridge clotting	High membrane pressure without the possibility of returning blood to the patient, with evidence of clot at the cartridge	Check vascular access flow before initiating HP Illuminate clot with spotlight Check access and return pressure alarm Forceful return of blood to patient, if possible
Circuit shattering	Circuit shaking during HP	Check vascular access Check patient volume status Fluid challenge

**Table 2 jcm-13-03360-t002:** The patients’ baseline characteristics before hemoperfusion.

Variables	All Cases (n = 27)	Phase I (n = 9)	Phase II (n = 18)	*p*-Value
Age (yrs)	61 (52, 67)	63 (53, 67)	58 (52, 67)	0.81
Female, n (%)	10 (37)	5 (56)	5 (28)	0.22
Body weight (kg)	70 (56, 83)	64 (56, 80)	75 (56, 83)	0.57
Height (cm)	160 (155, 170)	165 (156, 166)	159 (155, 170)	0.62
Body mass index (kg/m^2^)	25.4 (21.4, 31.1)	23.5 (21.4, 28.1)	27.2 (22.9, 31.1)	0.46
Diabetes mellitus	10 (37)	3 (33)	7 (39)	0.56
Hypertension	16 (59)	7 (78)	9 (50)	0.17
Dyslipidemia	4 (15)	2 (22)	2 (11)	0.41
Chronic obstructive pulmonary disease	4 (15)	1 (11)	3 (17)	0.59
Others	9 (33)	3 (33)	6 (33)	0.66
Body temperature (°C)	36.4 (36.0, 37.0)	36.0 (35.6, 36.4)	36.7 (36.4, 37.2)	<0.001
Heart rate (beats/min)	76 (62, 88)	62 (52, 76)	80 (72, 88)	0.044
Mean arterial pressure (mmHg)	89 (80, 103)	91 (83, 104)	88 (77, 102)	0.32
Respiratory rate (breaths/min)	24 (22, 28)	28 (24, 28)	24 (21, 28)	0.50
Pulse oximetry/fractional inspire oxygen	188 (115, 235)	190 (115, 235)	180 (119, 228)	0.97
National Early Warning Score 2	8 (7, 10)	10 (7, 10)	7 (6, 10)	0.24
Absolute lymphocyte count 10^3^ (/mm^3^)	677 (484, 1220)	694 (500, 1173)	636 (384, 1220)	0.60
D-dimer (ng/mL)	556 (423, 2526)	740 (481, 1734)	511 (376, 2526)	0.67
C-reactive protein (mg/L)	57 (39, 92)	88 (53, 134)	57 (16, 80)	0.15
Interleukin-6 (pg/mL) *	75 (29, 109)	79 (39, 86)	68 (29, 109)	0.93
Favipiravir, n (%)	23 (85)	7 (78)	16 (89)	0.41
Remdesivir, n (%)	26 (96)	8 (89)	18 (100)	0.33
Systemic corticosteroid (days)	11 (6, 15)	6 (5, 8)	12 (9, 16)	0.06
Tocilizumab, n (%)	9 (33)	1 (11)	8 (44)	0.09
Vasopressor, n (%)	14 (52)	6 (67)	8 (44)	0.25
Prone position, n (%)	17 (63)	5 (56)	12 (67)	0.44
High-flow nasal cannula, n (%)	22 (81)	7 (78)	15 (83)	0.55
Non-invasive ventilation, n (%)	9 (33)	0 (0, 0)	9 (50)	0.10
Mechanical ventilation, n (%)	22 (81)	7 (78)	15 (83)	0.55
Outcomes				
Intensive care unit mortality	8 (30)	3 (33)	5 (28)	0.55
Hospital mortality	9 (33)	3 (33)	6 (33)	0.66
Intensive care unit length of stay (days)	12 (7, 19)	10 (7, 14)	15 (8, 23)	0.22
Hospital length of stay (days)	13 (10, 23)	12 (9, 16)	15 (11, 24)	0.34

Continuous data are presented as median (IQR). * The interleukin-6 levels were aggregated from a subset of six cases during Phase II due to resource constraints in conducting cytokine analyses.

**Table 3 jcm-13-03360-t003:** Hemoperfusion-related outcomes and adverse events.

HP Characteristics	All Cases (n = 27)	Phase I (n = 9)	Phase II (n = 18)	*p*-Value
1st HP initiation from admission (days)	2 (1, 5)	1 (1, 2)	3 (2, 6)	0.08
Total HP sessions (sessions)	60	21	39	n/a
Median HP sessions (sessions/case)	2 (2, 3)	2 (2, 3)	2 (2, 3)	0.78
No. of HP sessions/case, n (%)				0.72
1	6 (22)	2 (22)	4 (22)	
2	10 (37)	3 (33)	7 (39)	
3	10 (37)	3 (33)	7 (39)	
4	1 (4)	1 (11)	0 (0)	
Total duration of HP operation (min)	480 (420, 720)	480 (420, 720)	480 (480, 720)	0.91
HP success rate per case, n (%)	22/27 (81)	6/9 (67)	16/18 (89)	0.19
HP success rate per session, n (%)	54/60 (90)	17/21 (81)	37/39 (95)	0.11
Total adverse events during HP, n	49	26	23	n/a
Median adverse events (events/case)	1 (0, 3)	3 (1, 4)	1 (0, 2)	0.014
Number of adverse events/cases, n (%)				0.039
0	7 (26)	0 (0)	7 (39)	
1–2	10 (37)	3 (33)	7 (39)	
3–6	10 (37)	6 (67)	4 (22)	
Adverse events, n (%)				
Shivering	3 (11)	2 (22)	1 (6)	0.25
Cardiac arrhythmia	5 (19)	3 (33)	2 (11)	0.19
Hypotension	7 (26)	4 (44)	3 (17)	0.14
Hypertension	4 (15)	2 (22)	2 (22)	0.41
Hypothermia	13 (48)	7 (78)	6 (33)	0.037
Cartridge clotting	5 (19)	1 (11)	4 (22)	0.45

Continuous data are presented as median (IQR). Abbreviation: HP, hemoperfusion and n/a, not applicable.

## Data Availability

Data are unavailable due to privacy or ethical restrictions.

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
