# Peer review of "Care Bundles to Improve Hemoperfusion Performance in Patients with Severe COVID-19: A Retrospective Study"

_jcm, 2024, doi:10.3390/jcm13123360_

Round 1

Reviewer 1 Report (Previous Reviewer 1)

Comments and Suggestions for Authors

Overall this is a study looking at the feasibility of using HP for cytokine storms from covid.  This study focuses on success of care bundles for HP and adverse events.  Its lacks any data or information on what effects the HP had and if it had any benefits.  

Comments on the Quality of English Language

English was fine no issues.

Author Response

Reviewer comment:

Overall, this is a study looking at the feasibility of using HP for cytokine storms from covid.  This study focuses on the success of care bundles for HP and adverse events.  Its lacks any data or information on what effects the HP had and if it had any benefits. 

Response

Thank you for taking the time to review our paper. The topic of HP benefits for COVID-19 is indeed intriguing. However, the study's restricted sample size presented a notable challenge in fully addressing this question.

Reviewer 2 Report (Previous Reviewer 2)

Comments and Suggestions for Authors

The authors have responded to all comment from previous submission and have made required changes to the manuscript. I have no further comments.

Author Response

Reviewer comment:

The authors have responded to all comments from previous submission and have made required changes to the manuscript. I have no further comments.

Response

Thank you for taking the time to review our paper. We appreciate that all previous responses have satisfied the reviewer.  

Reviewer 3 Report (New Reviewer)

Comments and Suggestions for Authors

Abstract:
- Although brief, the abstract leaves out important information about the procedures and findings. There should be more quantitative information on the results, statistical significance, and sample sizes provided.
- Grammatical mistake on line 16 - "Following the implementation of care bundles, there was a significant reduction in hypothermia (78% vs. 33%, p = 0.037), with an odds ratio of 0.14; 95% CI 0.02-0.91, p = 0.039 after adjusting for baseline BT." Rather than 0.039, the p-value should be 0.044.

Introduction
The introduction does an excellent job of setting the scene, but it falls short of emphasizing the information gaps that the current study seeks to fill. The goals and justification should be made more explicit.

- Some sentences are unduly complicated and lengthy. Readability will increase if the language is made simpler.

- Covid-19 influence several pathogenetic mechanisms as olfaction, dermatologic, respiratory distress and also otitis media. please discuss and cite doi:10.1007/s00405-021-06958-4

Methods:
- More information on the statistical analysis strategy is required, including which tests were employed for which outcomes, whether multiple testing adjustments were made, whether power analysis was carried out, etc.
- A more thorough description of the care bundle interventions used in Phase II should be provided. The changes could be summed up in a table.
Further details regarding the data gathering procedure are required. Were data extracted from medical records in the past? Who carried out the extraction? Were the study stages hidden from the researchers?

Results:
- The majority of the outcomes are statistical descriptions. Reporting the results that are clinically and statistically significant should be given more weight.
- The odds ratios should be given together with the confidence intervals.
Table 3's data need a more thorough explanation in the text; don't only say there were no notable differences. Analyze the outcomes.

Discussion:
- The discussion is not well-developed. More analysis of the results, comparisons with earlier research, study limitations, and practical implications are needed.
- Overstates the results considering the tiny sample size and the small number of statistically significant phase differences. It is important to recognize this as a restriction.
- Numerous grammatical mistakes throughout, including improper verb tenses, word choice, omitted words, etc. It has to be proofread thoroughly.

- discuss how after covid-19 office the regulation of office based and surgical room changed. However environmental consideration should be performed. cite doi:10.1093/bjs/znad092
- The manuscript does not adhere to the conventions of science. Sections should be enlarged and the organization changed.

Comments on the Quality of English Language

none

Author Response

Response to Reviewer 3 Comments

1. Summary

2. Questions for General Evaluation

Questions for General Evaluation

Reviewer’s Evaluation

Response and Revisions

Does the introduction provide sufficient background and include all relevant references?

Can be improved.

Thank you for your rating.

Are all the cited references relevant to the research?

Can be improved.

Thank you for your rating.

Is the research design appropriate?

Can be improved.

Thank you for your rating.

Are the methods adequately described?

Can be improved.

Thank you for your rating.

Are the results clearly presented?

Can be improved.

Thank you for your rating.

Are the conclusions supported by the results?

Can be improved.

Thank you for your rating.

3. Point-by-point response to Comments and Suggestions for Authors

Reviewer comment:

Abstract:
- Although brief, the abstract leaves out important information about the procedures and findings. There should be more quantitative information on the results, statistical significance, and sample sizes provided.

Response

Thank you for your insightful comments. We have revised the abstract in line with the reviewer's suggestions, particularly regarding the procedures of the care bundles. Due to the journal's recommendation to keep the abstract under 250 words, we have carefully refined the content to include the most essential information. The updated abstract is as follows:

Abstract: Background/Objectives: Hemoperfusion (HP) is employed to modulate cytokine storms in severe coronavirus disease 2019 (COVID-19) patients, requiring careful attention for success and safety. Therefore, we investigated whether our care bundles could enhance HP performance. Methods: We conducted a retrospective cohort study on adult patients (≥ 20 years old) with severe COVID-19 pneumonia. In the first wave (Phase I), we identified HP-related issues and addressed them with care bundles in the second wave (Phase II). The care bundles included early temperature control, precise hemodynamic monitoring, and clot prevention measures for the HP membrane. The HP success rate and associated adverse events (AEs) were assessed between the two phases. Results: The study included 60 HP (HA330) sessions from 27 cases (Phase I: n=21 sessions from 9 cases; Phase II: 39 from 18 cases). Patient characteristics and treatments for COVID-19 were similar, except for baseline body temperature (BT) and heart rate (HR). Phase II showed a higher success rate (67% vs. 89%, p = 0.19), although it did not reach statistical significance. Phase I significantly recorded a higher frequency of AEs (3 [IQR 1,4] events/case vs. 1 [IQR 0,2] events/case, p = 0.014). After implementing care bundles, hypothermia significantly decreased (78% vs. 33%, p = 0.037), with an adjusted odds ratio of 0.15; 95% CI 0.02-0.95, p = 0.044 for baseline BT. Conclusions: Further exploration with a larger sample size is required to establish the advantages of care bundles. However, the bundles implementation has significantly improved hypothermia prevention.

- Grammatical mistake on line 16

Response

Thank you for your comment. We have rewritten as suggested.

- "Following the implementation of care bundles, there was a significant reduction in hypothermia (78% vs. 33%, p = 0.037), with an odds ratio of 0.14; 95% CI 0.02-0.91, p = 0.039 after adjusting for baseline BT." Rather than 0.039, the p-value should be 0.044.

Response

Thank you for your comment. After careful review, we have confirmed that the original content is indeed accurate. We appreciate your attention to detail.

Introduction
The introduction does an excellent job of setting the scene, but it falls short of emphasizing the information gaps that the current study seeks to fill. The goals and justification should be made more explicit.

- Some sentences are unduly complicated and lengthy. Readability will increase if the language is made simpler.

Response

Thank you for your valuable feedback. We appreciate your suggestion and have rewritten the text to enhance readability as follows:

The coronavirus disease 2019 (COVID-19) pandemic ushered the world into a chaotic situation.[1,2] The emergence of this new contagious disease intensifies COVID-19 into becoming a burdensome disease. Numerous countries experienced a surge in the number of patients, resulting in a high rate of morbidity and mortality.[3-6]

Excessive production of inflammatory cytokines, such as interleukin-1 (IL-1), interleukin-6 (IL-6), and tumor necrotic factor alpha (TNF-alpha), significantly contributes to multi-organ dysfunction in COVID-19 patients.[7-10] Modulating the inflammatory response is crucial for promoting survival, such as corticosteroid,[11,12] IL-6 receptor antagonist,[13,14] and hemoperfusion (HP).[15,16] However, all of these treatments have distinct indications and levels of effectiveness.[7-9]

HP serves as a rescue therapy for severe COVID-19, countering cytokine storms caused by the virus.[17-19] However, it demands an expert team and carries peri-procedural related adverse events (AEs). During the first pandemic (Phase I), problems related to HP care were addressed in subsequent phases (Phase II) by implementing a care bundle, which aided nurses in conducting HP effectively. We investigated HP success rates and associated AEs before and after implementing care bundles in severe COVID-19 patients.

- Covid-19 influences several pathogenetic mechanisms as olfaction, dermatologic, respiratory distress and also otitis media. please discuss and cite doi:10.1007/s00405-021-06958-4

Response

Thank you for your thoughtful comment. Your diligence in this matter is greatly appreciated.

We are committed to ensuring that all references included in our work are directly relevant. Upon reviewing the article you provided, it appears that the study on the impact of the COVID-19 pandemic on otitis media (OM) may not align with the focus of our paper. Moreover, HP itself is not a treatment option for OM.

Methods:
- More information on the statistical analysis strategy is required, including which tests were employed for which outcomes, whether multiple testing adjustments were made, whether power analysis was carried out, etc.

Response

Thank you for your valuable comment. We have rewritten the statistical part as follows:

Univariable logistic regression analysis was utilized to illustrate the association of the care bundles, as the independent variable, with the dependent outcome. The dependent variables were put into the model one-by-one in a binary form (yes/no), which included the success rate and each AE. The results were reported as the odds ratio (OR) and 95% confidence interval (95%CI).

Given the constraints of a limited sample size, we focused our selection on pertinent variables that exhibited differences at baseline and were related to the outcome of interest for refining the regression model such as adjusting body temperature in the model where the outcome was hypothermia.

- A more thorough description of the care bundle interventions used in Phase II should be provided. The changes could be summed up in a table.

Response

Thank you for your comment. The original content already included this information in the third collum of Table 1, as follows:

Table 1. Adverse Events and Care Bundles for Hemoperfusion.

  Adverse events

Definition

Bundles of care

Shivering

The presence of body shaking or teeth chattering

Give paracetamol, opioids, or sedative

Check for membrane allergy

Provide warm blanket if hypothermia

Cardiac arrhythmia

New onset of atrial fibrillation/flutter, premature ventilation, among others

Ask about chest pain

Check volume status

Check electrolytes

Reduce blood flow rate

Give anti-arrhythmic drugs

Hypotension

Mean arterial pressure ≤ 65 mm Hg

Decrease preset temp to 36.5°C

Check the patient volume status

Fluid challenge if indicated

Reduce blood flow rate

Hypertension

Blood pressure > 140/90 mmHg

Manage pain/agitation

Give anti-hypertensive drugs

Hypothermia

Body temperature < 36.0°C

Check patient temp before HP

Increase HP preset temp

Provide warm blanket

Cartridge clotting

High membrane pressure without the possibility of returning blood to the patient with the evidence of clot at the cartridge

Check vascular access flow before initiating HP

Illuminate a clot with a spotlight

Check access and return pressure alarm

Forceful return of blood to the patient if possible

Circuit shattering

Circuit shaking during hemoperfusion

Check vascular access

Check the patient volume status

Fluid challenge

- Further details regarding the data gathering procedure are required. Were data extracted from medical records in the past? Who carried out the extraction? Were the study stages hidden from the researchers?

Response

All information was previously sourced from the Electronic Medical Records. SM, one of our researchers, retrieved the information.

The patient care and bundle implementation occurred during the COVID pandemic. However, the researchers subsequently gathered the information and conducted the statistical analysis at a later stage.

We have included additional information as follows:

The data were retrieved from the Electronic Medical Records by the first author, SM. The information included the patients’ demographics, pre-existing comorbidities, vital signs, severity of illness, and laboratory test results at the initial phase of HP therapy. Throughout the patient’s ICU stay, we documented treatments and types of respiratory support.

Results:
- The majority of the outcomes are statistical descriptions. Reporting the results that are clinically and statistically significant should be given more weight.

Response

Thank you for your comment. We have rewritten more details as follows:

We included all patients with severe COVID-19 who underwent HP at our center. Twenty-seven cases were involved in the study (Phase I: n=9 and Phase II: n=18). Table 2 summarizes the baseline characteristics of the patients. When comparing Phase I and Phase II, there were no significant differences in patients’ demographics, including age and gender, with median values of 63 [IQR 53,67] years vs. 58 [IQR 52,67] years, p = 0.81 and females comprising 56% vs. 28%, p = 0.22, respectively.  

Moreover, pre-existing comorbidities and vital signs were also comparable between phases, except for body temperature (BT) and heart rate (HR). Both BT and HR were significantly lower in Phase I than in Phase II, with the values of 36.0 [IQR 35.6, 36.4] °C vs. 36.7 [IQR 36.4, 37.2] °C, p < 0.001 and 62 [IQR 52,76] bpm vs. 80 [IQR 72,88] bpm, p = 0.044, respectively.

     Additionally, there were no discernible variations observed in the results of laboratory investigations, including absolute lymphocyte count, D-dimer, and C-reactive protein, which were not different between phases (all p > 0.005). Table 2 further provides information about treatment administered during the ICU admission. Of note, there were no significant differences in the treatments, type of respiratory support, and patient outcomes between the two phases (all p > 0.05).

- The odds ratios should be given together with the confidence intervals.

Response

Thank you for your comment. We think that the original content already included this information.

- Table 3's data need a more thorough explanation in the text; don't only say there were no notable differences. Analyze the outcomes.

Response

Thank you for your comment. We have rewritten more details regarding Table 3 and also incorporated the denominator into the HP success rate outcome in the table to enhance clarity and comprehension as follows:

All HP-related information is summarized in Table 3. The time to first HP initiation from admission was not different between phases (1 [IQR 1,2] days vs. 3 [IQR2,6] days, p = 0.08). In total, there were 60 HP sessions/27 cases (Phase I: 21 sessions/9 cases and Phase II: 39 sessions/18 cases), with no significant differences in the median HP sessions per case and the total duration of HP operation between phases (2 [IQR 2,3] sessions vs. 2 [IQR 2,3] sessions, p = 0.78 and 480 [IQR 420,720] minutes vs. 480 [IQR 480,720] minutes, p = 0.91).

The success rate of HP per case, calculated from the number of cases that totally completed a 4-hour HP session each divided by the number of cases from each phase, was slightly higher in Phase II (Table 3). However, it did not reach statistical significance (67% vs. 89%, p = 0.19). The success rate remained the same when considered per session, calculated from the number of sessions that completed a 4-hour session divided by the number of sessions from each phase (81% vs. 95%, p = 0.11).

The total number of AEs was 49 events (Phase I: 26 events and Phase II: 23 events). The median number of AEs per case was significantly higher in Phase I than in Phase II (3 [IQR 1,4] events/case vs. 1 [IQR 0,2] events/case, p = 0.014). The details regarding the number of AEs per case differed between phases (p = 0.039), as summarized in Table 3.

While the incidence of shivering, cardiac arrhythmia, hypotension, and hypertension was not significantly different (all p > 0.05), hypothermia exhibited a statistically significant reduction (78% vs. 33%, p = 0.037), with an OR of 0.14; 95% CI 0.02-0.91, p = 0.039 (Figure 1). Adjusting for different baseline BT values, the risk of hypothermia remained significantly reduced in Phase II, with an OR of 0.15; 95% CI 0.02-0.95, p = 0.044 (Figure 1).

Table 3. Hemoperfusion-related outcomes and adverse events.

HP Characteristics

All cases

(n=27)

Phase I

(n=9)

Phase II

(n=18)

P-value

1st HP initiation from admission (days)

2 (1,5)

1 (1,2)

3 (2,6)

0.08

Total HP sessions (sessions)

60

21

39

n/a

Median HP sessions (sessions/case)

2 (2,3)

2 (2,3)

2 (2,3)

0.78

No. of HP sessions/case, n (%)

0.72

1

6 (22)

2 (22)

4 (22)

2

10 (37)

3 (33)

7 (39)

3

10 (37)

3 (33)

7 (39)

4

1 (4)

1 (11)

0 (0)

Total duration of HP operation (min)

480 (420,720)

480 (420,720)

480 (480,720)

0.91

HP success rate per case, n (%)

22/27 (81)

6/9 (67)

16/18 (89)

0.19

HP success rate per session, n (%)

54/60 (90)

17/21 (81)

37/39 (95)

0.11

Total adverse events during HP, n

49

26

23

n/a

Median adverse events (events/case)

1 (0,3)

3 (1,4)

1 (0,2)

0.014

Number of adverse events/cases, n (%)

0.039

0

7 (26)

0 (0)

7 (39)

1-2

10 (37)

3 (33)

7 (39)

3-6

10 (37)

6 (67)

4 (22)

Adverse events, n (%)*

Shivering

3 (11)

2 (22)

1 (6)

0.25

Cardiac arrhythmia

5 (19)

3 (33)

2 (11)

0.19

Hypotension

7 (26)

4 (44)

3 (17)

0.14

Hypertension

4 (15)

2 (22)

2 (22)

0.41

Hypothermia

13 (48)

7 (78)

6 (33)

0.037

Cartridge clotting

5 (19)

1 (11)

4 (22)

0.45

Continuous data are presented as median (IQR). Abbreviation: HP, hemoperfusion.

Discussion:
- The discussion is not well-developed. More analysis of the results, comparisons with earlier research, study limitations, and practical implications are needed.

Response

Thank you for your input. We acknowledge the absence of comprehensive data on optimal HP operation practices, which limits our ability to make direct comparisons. Nevertheless, we have addressed the constraints related to data comparison and implications in a subsequent paragraph.

Although several studies demonstrated the advantages of HP for severe COVID-19 patients, [17,19,23-29] a notable deficiency exists in the available evidence to support best practices for enhancing HP performance. In addition, the overwhelming number of patients with severe COVID-19 led to a widespread shortage of healthcare professionals. In response, some centers eventually enlisted multidisciplinary healthcare workers who may not be specialized in critical care medicine. Our center encountered this challenge as well. Therefore, the execution of HP operations could occasionally present challenges. The guidance from experienced nurses, coupled with comprehensive care bundles encompassing all essential elements, became indispensable. It is worth noting that this study originates from a resource-limited setting where the implementation of HP can be somewhat financially burdensome. Given that the success rate of HP operations could potentially influence the survival of severe COVID-19 patients, it becomes imperative to develop a strategy for enhancing HP's performance.  

- Overstates the results considering the tiny sample size and the small number of statistically significant phase differences. It is important to recognize this as a restriction.

Response

We appreciate your insights. We recognize that the constrained sample size impacts the precision and applicability of our study findings. We have duly acknowledged this concern as the top priority in the limitations section outlined below:

Our study had some limitations. Firstly, the small number of patients with the study hypothesis centered on the care bundle to enhance HP performance impeded a comprehensive assessment of HP benefits, including the success rates and other efficacies such as disease progression, intubation, or mortality rates. Further investigations with larger sample sizes could provide more insights to assess the impact of the care bundles in cooperation with standardized treatment protocol, including type and duration of corticosteroid or the use of IL-6 receptor inhibitor, in enhancing patients’ survival outcomes.

- Numerous grammatical mistakes throughout, including improper verb tenses, word choice, omitted words, etc. It has to be proofread thoroughly.

Response

Thank you for your input. We have consulted a native English speaker to proofread the manuscript as suggested.

- Discuss how after covid-19 office the regulation of office based and surgical room changed. However environmental consideration should be performed. cite doi:10.1093/bjs/znad092

Response

Upon reviewing the article you provided, it appears that this study represents a crucial stride towards greener healthcare practices, offering actionable interventions applicable across diverse economic landscapes. Nevertheless, this article is not relevant to our study.

- The manuscript does not adhere to the conventions of science. Sections should be enlarged and the organization changed.

Response

We sincerely appreciate the time you dedicated to reviewing our paper and value all the insightful suggestions provided by the reviewer. However, this comment could benefit from further specificity. Providing more detailed feedback would greatly assist us in enhancing the manuscript to align with the high standards of scientific reporting.

Round 2

Reviewer 1 Report (Previous Reviewer 1)

Comments and Suggestions for Authors

The paper fixes most of the problems in the first draft.  How ever it shows that a bundle cared for HP is better then no bundle.  However it lacks any useful information if HP is effective for COVID and if HP is better then standard treatment.  

Author Response

Research article “Care Bundles to Improve Hemoperfusion Performance in Patients with Severe COVID-19: A Retrospective Study

Journal: JCM (ISSN 2077-0383)

Manuscript ID: jcm-3015841

Response to Reviewer 1 Comments

Reviewer comment:

The paper fixes most of the problems in the first draft.  How ever it shows that a bundle cared for HP is better than no bundle.  However, it lacks any useful information if HP is effective for COVID and if HP is better than standard treatment. 

Response

Thank you for taking the time and effort to review our paper. The topic of HP benefits for COVID-19 is indeed intriguing.

Accordingly, our study cannot overclaim the conclusion that HP is the best option for severe COVID-19. However, we found that the levels of IL-6, an important surrogate marker, were markedly reduced following HP treatment, as shown in the table.

Phases

Before HP

After HP

P-value

All phases

75 (29,109)

25 (9,60)

<0.001

Phase I

79 (39, 86)

35 (9,60)

0.004

Phase II

68 (29,109)

25 (13,27)

0.03

Therefore, we have included this information in the discussion section:

Our study had some limitations. Firstly, the small number of patients with the study hypothesis centered on the care bundles to enhance HP performance impeded a comprehensive assessment of HP benefits, including the success rates and other efficacies such as disease progression, intubation, or mortality rates. Further investigations with larger sample sizes could provide more insights to assess the impact of the care bundles in cooperation with standardized treatment protocol, including type and duration of corticosteroid or the use of IL-6 receptor inhibitor, in enhancing patients’ survival outcomes. However, in terms of efficacy we observed a significant reduction in IL-6, an important surrogate marker for disease severity, following HP treatment. In all phases, the levels of IL-6 decreased from 75 (IQR 29,109) pg/mL to 25 (IQR 9,60) pg/mL, p < 0.001. This reduction was consistent across both phases. In phase I, IL-6 levels dropped from 79 (IQR 39,86) pg/mL to 35 (IQR 9,60) pg/mL, p < 0.005). In phase II, IL-6 levels dropped from 68 (IQR 29,109) pg/mL to 25 (IQR 13,27) pg/mL, p = 0.03.

Additionally, we have summarized and highlighted valuable insights on HP for COVID-19 from previous studies as follows:

Several studies have explored the application of HP in severe COVID-19 patients, utilizing various cartridge types, including HA330, [17,19] HA380, [23] HA230 in combination with HA280, [24] CytoSorb, [25-27], and oXiris. [28,29). While HP has shown promise in improving SpO2 levels and reducing inflammatory cytokines, it is important to note that most of these studies have been constrained by their small sample sizes and case series designs.[23-26,28,29]

One of the largest investigations of CytoSorb involved the CYCOV trial, [27] which revealed that the level of IL-6 at 72 hours after ECMO initiation did not significantly differ

between the HP+ECMO group (n=17) vs. the ECMO alone group (n=17). However, the 30-day survival was significantly lower in the HP+ECMO group (18% vs. 76%, p = 0.002). Caution is required when considering HP during the early phase of ECMO therapy, as HP can also remove anti-inflammatory cytokines that might potentially contribute to this adverse outcome.

To our knowledge, one large retrospective study (n=128) compared the use of HP with HA330 (n=46) and CytoSorb (n=9) to matched controlled patients (n=73).[19] This study showed a lower ICU mortality in the HP group than in the control group (67% vs. 89%, p = 0.002).[19] The HP group also exhibited more favorable outcomes in terms of a shorter ICU length of stay, greater improvements in SpO2, and more reduction in PaCO2 when compared to the control group.[19] However, it is worth noting that the mortality rate in this study was somewhat higher than in our study (30%). Nonetheless, intubation rates were similar, with approximately three-fourths of patients being intubated.   

Another study employing HA330 was conducted by Surasit K et al., where HP was performed for three or more sessions (n=15) compared to no HP or less than three sessions of HP (n=14) .[17] The results indicated that patients who received three or more HP sessions achieved a reduction in organ dysfunction (as measured by the SOFA score), a decrease in pulmonary infiltration on chest X-ray, and lower CRP levels. Furthermore, the HP group outperformed the control group in terms of ICU mortality and 28-day mortality (13.3% vs. 92.9%, p < 0.001 and 13% vs. 86%, p < 0.001, respectively).

Unfortunately, we were unable to demonstrate how the efficacy of HP compares to standard treatment, as our study did not include data from patients who received only the standard therapy, in accordance with the scope of our research.

This manuscript is a resubmission of an earlier submission. The following is a list of the peer review reports and author responses from that submission.

Round 1

Reviewer 1 Report

Comments and Suggestions for Authors

The paper is a small retrospective study in a resource poor hospital looking at the use of Hemoperfusion for COVID-19. 

It compares wave 1 and wave 2 and shows how changes made to the protocol limited the AE from HP for these patients. 

However, the study has no control and focuses mostly on limiting AE. 

It does not cover any data or discussion on improvements.

Author Response

Response to Reviewer 1 Comments

1. Summary

2. Questions for General Evaluation

Questions for General Evaluation

Reviewer’s Evaluation

Response and Revisions

Does the introduction provide sufficient background and include all relevant references?

Yes

Thank you for your rating.

Are all the cited references relevant to the research?

Yes

Thank you for your rating.

Is the research design appropriate?

Yes

Thank you for your rating.

Are the methods adequately described?

Yes

Thank you for your rating.

Are the results clearly presented?

Must be improved

We have rewritten as the review suggestion.

Are the conclusions supported by the results?

Can be improved

We have reached our conclusions. Should you have any specific recommendations or insights, we would be delighted to incorporate them.

3. Point-by-point response to Comments and Suggestions for Authors

Reviewer comment:

The paper is a small retrospective study in a resource poor hospital looking at the use of Hemoperfusion for COVID-19. It compares wave 1 and wave 2 and shows how changes made to the protocol limited the AE from HP for these patients. However, the study has no control and focuses mostly on limiting AE. It does not cover any data or discussion on improvements.

Response

Thank you for taking the time to review our paper. We appreciate your feedback.

Our study indeed focuses on the utilization of Hemoperfusion for COVID-19 patients in a resource-limited hospital setting, presenting a comparative analysis between wave 1 and wave 2 of the pandemic. We acknowledge the importance of controlled studies, and while our retrospective design may have limitations in this regard, we aimed to shed light on the impact of protocol modifications on adverse events (AE) associated with Hemoperfusion.

Regarding your observation that our study primarily emphasizes limiting AE and lacks discussion on improvements, we acknowledge this limitation. We agree that discussing potential improvements and future directions would add valuable insights. In our future research endeavors, we will strive to incorporate a more comprehensive analysis encompassing both AE mitigation strategies and potential enhancements to the treatment protocol.

Please look at the discussion section:

Page 7 Paragraph II-V, Page 8 Paragraph V-VI, and Page 9 Paragraph I-II 

Reviewer 2 Report

Comments and Suggestions for Authors

In this manuscript authors present the effect of care bundle on hemoperfusion performance i severe COVID-19 patients. Article is well written and structured. Methods and  results are clearly presented and discussion follows the results. Limitations of the study are described and presented. There are few things I would like the authors to address.

1. Please add study type in the title.

2. Abstract:  please rephrase „Phase II showed a higher HP success rate”, “Phase I recorded a higher frequency…” It should say: a higher success rate was recorded in Phase II… Higher frequency was recorded in phase I…"

3. Methods: did you report the study in accordance with relevant reporting guidelines? ( https://www.equator-network.org/reporting-guidelines/strobe/)

4. Methods: In inclusion criteria it says patients who were > 20 years, while in abstract it says that included patients were > 18 years. Please correct.

5. Methods: please clearly define Phase I and Phase II in terms of time period. Is it possible to set a clear time limit between the two periods?

6. In rows 136-137 it says: 

"We employed regular monitoring of the patient’s vital   

signs before the HP procedure and at 5, 15, 30, 45, 60, 120, 180, and 240 minutes thereafter". Does this refer to "Phase II" only? in that case, monitoring of vital signs was not regularly employed during "Phase I"?

7. Results: Please add total number of patients treated during study period in your center. Also, please add total number of patients analyzed in the abstract in addition to total number of procedures analyzed.

8. In table 2., CRP levels are shown in mg/L while in Methods section CRP levels are presented in mg/dL. These units should be uniformly presented through the manuscript.

9. In the Discussion section, authors could refer to the differences in some of the patients characteristics from Table 2 and their possible effect on study outcomes. For example, altough there was no statistically significant difference, number of patients receiving corticosteroids and tocilizumab is markedly higher in Phase II group compared to patients from Phase I group (median of corticosteroid th is two times larges and number of patients receiving tocilizumab iz 4 times higher in Phase II patients). Is it possible that these factors could influence the outcomes? Also, the time of initiation oh HP therapy was delayed in Phase II patients for two days compared to patients from Phase I. Could this be also one of the possible factors that influence outcomes?

10. Discussion: Since the main outcome of this study was the influence of care bundles on HP therapy, there should be more emphasis on this in discussion.

Author Response

Response to Reviewer 2 Comments

1. Summary

2. Questions for General Evaluation

Questions for General Evaluation

Reviewer’s Evaluation

Response and Revisions

Does the introduction provide sufficient background and include all relevant references?

Can be improved

We have rewritten it as the review suggestion.

Page 2 Paragrap I Lines 48-51.

Are all the cited references relevant to the research?

Yes

Thank you for your rating.

Is the research design appropriate?

Can be improved

We appreciate your rating. We acknowledge that there may be opportunities to enhance the study. If you have any specific recommendations for improving the study design, we are open to making adjustments in line with the recommendations.

Are the methods adequately described?

Yes

Thank you for your rating.

Are the results clearly presented?

Yes

Thank you for your rating.

Are the conclusions supported by the results?

Yes

Thank you for your rating.

3. Point-by-point response to Comments and Suggestions for Authors

Reviewer comment:

In this manuscript authors present the effect of care bundle on hemoperfusion performance i severe COVID-19 patients. Article is well written and structured. Methods and results are clearly presented and discussion follows the results. Limitations of the study are described and presented. There are few things I would like the authors to address.

Response

First, I would like to thank you for taking the time to review our paper. We appreciate your feedback. The point-by-point response are as follows:

1. Please add study type in the title.

Response

Thank you for the suggestion, we have added “A Retrospective Study” at the end of the title.

Please look at the title.

2. Abstract: please rephrase, Phase II showed a higher HP success rate”, “Phase I recorded a higher frequency…” It should say: a higher success rate was recorded in Phase II… Higher frequency was recorded in phase I…"

Response

Thank you for the suggestion, we have corrected as the reviewer’s suggestion.

Please look at Page 1 Line 23-24 and Line 25

3. Methods: did you report the study in accordance with relevant reporting guidelines?

(https://www.equatornetwork.org/reporting-guidelines/strobe/)

Response

Please look at the attached STROBE checklist.

4. Methods: In inclusion criteria it says patients who were > 20 years, while in abstract it says that included patients were > 18 years. Please correct.

Response

Thank you for the comment. We have corrected the age of participant of 20 or greater throughout the manuscript.

Please look at Page 1 Line 18.

5. Methods: please clearly define Phase I and Phase II in terms of time period. Is it possible to set a clear time limit between the two periods?

Response

Thank you for the comment. To provide further clarification, we have added “(Phase I: 12 April 2021 to 19 May 2021 and Phase II: 14 July 2021 to 27 January 2022, with a total number of patients in each phase of 63 and 180 cases, respectively).” to the method section.

Please look at Page 2 Lines 57-58.

6. In rows 136-137 it says: "We employed regular monitoring of the patient’s vital signs before the HP procedure and at 5, 15, 30, 45, 60, 120, 180, and 240 minutes thereafter". Does this refer to "Phase II" only? in that case, monitoring of vital signs was not regularly employed during "Phase I"?

Response

We have rewritten for further clarification as follows:

“We conducted frequent monitoring of the patient's vital signs before the HP procedure and at intervals of 5, 15, 30, 45, 60, 120, 180, and 240 minutes thereafter. This standardized monitoring of vital signs before and after the initiation of HP was not fully established in Phase I. There was an understanding that vital signs were observed every fifteen minutes during the initial hour of HP, and subsequently every sixty minutes, in accordance with the tailored nursing approaches for each unit. Nevertheless, to foster uniformity in nursing practices, efforts were made to integrate these practices into a formalized protocol or agreement, particularly in Phase II.”

Please look at Page 8 Last paragraph.

7. Results: Please add total number of patients treated during study period in your center. Also, please add total number of patients analyzed in the abstract in addition to total number of procedures analyzed.

Response

We have rewritten as follows:

The study included 60 HP (HA330) sessions from 27 cases (Phase I: n=21 session from 9 cases; Phase II: 39 from 18 cases).

Please look at Page 1 Lines 22-23.

8. In table 2., CRP levels are shown in mg/L while in Methods section CRP levels are presented in mg/dL. These units should be uniformly presented through the manuscript.

Response

Thank you for the comment. In our hospital the unit of CRP is mg/L. We have made the adjustment to ensure consistency regarding CRP units across the manuscript.

Please look at Page 3 Line 121.

9. In the Discussion section, authors could refer to the differences in some of the patients characteristics from Table 2 and their possible effect on study outcomes. For example, altough there was no statistically significant difference, number of patients receiving corticosteroids and tocilizumab is markedly higher in Phase II group compared to patients from Phase I group (median of corticosteroid th is two times larges and number of patients receiving tocilizumab iz 4 times higher in Phase II patients). Is it possible that these factors could influence the outcomes? Also, the time of initiation of HP therapy was delayed in Phase II patients for two days compared to patients from Phase I. Could this be also one of the possible factors that influence outcomes?

Response

Thank you for the comment.

We have rewritten the discussion as follows:

Interestingly, there was a slightly higher number of patients undergoing HP therapy in Phase I compared to Phase II, though lacking statistical significance. The proportion of patients receiving HP during these two phases was 14.3% (9/63 cases) vs. 10.0% (18/180 cases), respectively, with a p-value of 0.36. One contributing factor to the reduction in the utilization of HP treatment in Phase II pertained to the necessity of vascular access and specialized nurse support for continuous bedside HP operation lasting for at least four hours. This additional complexity rendered HP more intricate than mere medication administration.

Moreover, as time progressed, more knowledge supporting best practices for treating the COVID-19 pandemic continued to evolve. We noticed some disparities in patient treatment between the two phases. The administration of systemic corticosteroids showed a tendency towards a longer duration in Phase II (6 [IQR 5,8] days vs. 12 [IQR 9,16] days, p = 0.06). Additionally, the usage of Tocilizumab, an IL-6 receptor inhibitor, was more prevalent in Phase II (11% vs. 44%, p=0.09).

Nonetheless, there were a total of 60 HP sessions in our study (Phase I: 21 sessions and Phase II: 39 sessions). The comparative success rate of HP between Phase I and Phase II showed no statistical difference when adjusted by case and by sessions (67% vs. 89%, p=0.19 and 81% vs. 95%, p=0.11, respectively). Remarkably, the introduction of the HP care bundle led to a substantial decrease in the median number of adverse events (3 [IQR 1,4] events/case vs. 1 [IQR 0,2] events/case, p=0.014). While shivering, cardiac arrhythmia, hypotension, and hypertension did not exhibit significant differences, hypothermia demonstrated a statistically significant reduction (78% vs. 33%, p = 0.037), with an OR 0.15; 95% CI 0.02-0.95, p = 0.044 when adjusted for baseline body temperature. This can be implied that the implementation of the care bundle during HP can effectively mitigate the occurrence of hypothermia.

Although the ICU mortality rate was marginally lower in Phase II, there was no statistically significant difference between the two phases (33% vs. 28%, p=0.55). One plausible explanation could be the slightly delayed initiation of HP compared to Phase I, with the median time to first HP initiation from ICU admission being 1 (IQR 1,2) day vs. 3 (IQR 2,6) days, p=0.008. The impact of a delay in HP initiation could potentially influence the efficacy of HP. Exploring the effectiveness of cytokine modulation through a combination of early HP coupled with cytokine reduction medication could offer valuable insights for future research. On the other hand, investigating the efficacy by comparing medications alone to early HP could also present another area of interest.

Please look at Page 7 Paragraph II-V.

10. Discussion: Since the main outcome of this study was the influence of care bundles on HP therapy, there should be more emphasis on this in discussion.

Response

Thank you for the comment.

We have rewritten the discussion as follows:

Hypothermia or suboptimal thermal regulation represents a noteworthy challenge in the context of extracorporeal organ support. Studies indicated that hypothermia af-fected nearly half of patients undergoing continuous renal replacement therapy (CRRT).[30,31] This condition demands attention due to its potential to exacerbate pa-tient thermal instability, thereby heightening susceptibility to sepsis, precipitating chills, and inducing arrhythmias and hemodynamic instability. Furthermore, individuals encountering hypothermia during CRRT were confronted with a notable increase in mortality risk, with rates reaching up to 60%.[31] It is anticipated that the incidence of hypothermia will remain consistent for HP. Although certain centers may exhibit a heightened susceptibility to hypothermia, it is important to note that some HP machines, including those in our center, lack integrated warming capabilities. Therefore, a protocol to monitor patient body temperature is necessitated to prevent this complication.

We conducted frequent monitoring of the patient's vital signs before the HP procedure and at intervals of 5, 15, 30, 45, 60, 120, 180, and 240 minutes thereafter. This standardized monitoring of vital signs before and after the initiation of HP was not fully established in Phase I. There was an understanding that vital signs were observed every fifteen minutes during the initial hour of HP, and subsequently every sixty minutes, in accordance with the tailored nursing approaches for each unit. Nevertheless, to foster uniformity in nursing practices, efforts were made to integrate these practices into a formalized protocol or agreement, particularly in Phase II. Regular monitoring of vital signs prompted nursing action during HP. Furthermore, the com-prehensive bundle facilitated timely physician consultation in cases where initial management falls short.

An additional issue that impacted the effectiveness of HP was the formation of clots within the membrane. Furthermore, blood loss of approximately 285 mL (HA330 = 185 mL and circuit = 100 mL) may occur during circuit disposal. Thus, it is advisable to promptly address any early indications of membrane clotting. This can be achieved by regularly checking vascular access, adjusting patient positioning, closely monitoring for any increase in transmembrane pressure (TMP) and examining clot size further when TPM rises. In advanced stages where machine operation is compromised due to clotting, a more vigorous blood return is recommended to minimize blood loss whenever feasible.

Please look at Page 8 Paragraph V-VI and Page 9 Paragraph I-II.
